# Magnesium ions improve vasomotor function in exhausted rats

**Dan Wang, Zong-Xiang Li, Dong-Mou Jiang, Yan-Zhong Liu, Xin Wang, Yi-Ping Liu***

Provincial University Key Laboratory of Sport and Health Science, School of Physical Education and Sport Sciences, Fujian Normal University, Fuzhou, China

* ypliu1966@fjnu.edu.cn

**Data Availability Statement:** All relevant data are within the paper and its Supporting Information files.

**Funding:** This work was supported by grant the China Postdoctoral Science Foundation

## Abstract

To observe the effect of magnesium ion on vascular function in rats after long-term exhaustive exercise. Forty male SD rats were divided into two groups, the control group (CON group, n = 20) and the exhaustive exercise group (EEE group, n = 20). Exhausted rats performed 1W adaptive swimming exercise (6 times/W, 15min/time), and then followed by 3W formal exhaustive exercise intervention. Hematoxylin and eosin (HE) staining was used to detect the morphological changes of rat thoracic aorta. The contents of interleukin-1 β (IL-1β) and tumor necrosis factor–α (TNF-α) in serum of rats were determined by enzyme-linked immunosorbent assay (ELISA), and the contents of malondialdehyde (MDA), reactive oxygen species (ROS), nitric oxide (NO) and endothelin 1 (ET-1) in serum of rats were determined by biochemical kit. Vascular ring test detects vascular function. Compared with the CON group, the smooth muscle layer of the EEE group became thicker, the cell arrangement was disordered, and the integrity of endothelial cells was destroyed; the serum $Mg^{2+}$ in EEE group was decreased; the serum levels of IL-1β, TNF-α, MDA and ROS in EEE group were significantly higher than those in the CON group ($P$ are all less than 0.05); the serum NO content in EEE group was significantly decreased, and the ratio of NO/ET-1 was significantly decreased. In the exhaustion group, the vasoconstriction response to KCl was increased, and the relaxation response to Ach was weakened, while 4.8mM $Mg^{2+}$ could significantly improve this phenomenon ($P$ are all less than 0.01). The damage of vascular morphology and function in rats after exhaustion exercise may be related to the significant increase of serum IL-1β, TNF-α, ROS, MDA and ET-1/NO ratio in rats after exhaustion exercise, while $Mg^{2+}$ can significantly improve the vasomotor function of rats after exhaustion exercise.

## Introduction

It is widely accepted that physical activity can promote health and reduce the risk of cardiovascular disease, cancer, diabetes and other chronic diseases, and there is a strong inverse relationship between physical fitness and mortality in humans [1, 2]. However, if the exercise intensity and exercise time exceed a certain limit, the beneficial effect of exercise may be lost, and generate a large amount of ROS through the electron transport chain, which will lead to

(2021M700782) and the Fund for Social Science Foundation, Fujian, China (FJ2021B138).

**Competing interests:** The authors have declared that no competing interests exist.

the oxidation-reduction instability of the body [3]. In addition, exhaustive exercise also triggers the release of a large number of pro-inflammatory factors by the leukocytes of the body, triggering chronic inflammation, and eventually causing muscle or tissue damage [4–6]. It has also been reported that exhaustive exercise led to significant impairment of ventricular systolic and diastolic function in rats [7]. Studies also have shown that exhaustive exercise leads to the injury of arterial morphology and function [8] and accompanied by the decrease of arterial compliance [9]. Therefore, preventing the damage of vascular function caused by exhaustive exercise plays an important role in maintaining the normal function of the cardiovascular system.

Magnesium ion ($Mg^{2+}$) is an abundant intracellular divalent cation, which participates in the metabolism and redox balance regulation of all tissues in the body [10]. $Mg^{2+}$ can also regulate the immune function, acting on cells of the innate and adaptive immune system, and regulate the inflammatory state of the body [11]. Magnesium also plays vital role in many biochemical, physiological and cellular processes involving in regulating cardiovascular functions [12]. It regulates vascular tone by changing the vascular actions to vasodilator and vasoactive agonists, and it affects endothelial function by modulating vasodilatation [13]. Accumulating researches have shown that maintenance of $[Mg^{2+}]_i$ can reduce the oxidative stress, inflammation and vascular remodeling [14, 15]. Studies have shown that $Mg^{2+}$ supplementation can improve the vasomotor function in rats with pulmonary hypertension [13], but the research on whether $Mg^{2+}$ has a protective effect on vascular function damage caused by exhaustive exercise has not been reported yet. In this context, we hypothesized that exhaustive exercise could damage vascular function by reducing magnesium ions, increasing oxidative stress and inflammation, and high magnesium can improve vascular function of rats after exhaustive exercise. Therefore, the purpose of this study is to investigate the effect of $Mg^{2+}$ on vascular function damage caused by exhaustive exercise and provide a theoretical support for magnesium ions to improve vascular function in sports.

## Materials and methods

### Animals

Eight-week-old adult male Sprague–Dawley (SD) rats, weight 200 ± 10g (means ± SEM), used in this study were purchased from Wushi Experimental Animal Supply (Fuzhou, China). Rats were placed under standard controlled environmental conditions (temperatures: 24–25°C, humidity: 45–55%, 12 h light: dark cycle), and fed a standard laboratory rat diet ad libitum with free access to water, and after one week of acclimatization, animals were randomly divided into two different experimental groups (20 animals/group): (1) Control group (CON), (2) Exhaustive exercise group (EEE). All experimental protocols conducted on the rats were approved by the Institutional Animal Care and Use Committee, Fujian Normal University (approval No.: 20210037). After the material collection, the experimental rats were euthanized by cervical dislocation method.

### Training protocol

Rats were allowed to swim adaptively for 1W before any exercise took place. The rats were familiarized with swimming in an apparatus holding no less than a water depth of 60cm for 15min daily at 8:30 am for 6 days/week during the first week. Then, the rats were subjected to a formal swim to exhaustive in deep water tanks (1.0 m in diameter and 0.7 m deep) with a water temperature of 34–36°C at 8:30 am for 6 days/week. This water temperature has previously been shown to be suitable for swimming [16], and the rats did not experience any cardiovascular or other side effects that could affect their performance [17]. A constant load

equivalent to 5% of the rat's body weight was adjusted at the rat's tail to achieve continuous swimming [18]. The rats were continuously monitored, and fatigue points are observed visually. When the rat failed to rise its nose out of the water for inhaling within 10s, and the righting reflex could not be performed after turning over when it was placed on a flat surface [19, 20], it was determined to be in a state of exhaustion and the exercise was terminated. The average exercise time was about 3 hours for 4-week [18, 20, 21]. Swimming was chosen because it is a different form of exercise than full-body treadmill running, which does not cause limited muscle damage [22]. Therefore, any effect of swimming on oxidative stress cannot be attributed to muscle damage, which increases the production of reactive species [23].

### Detection of serum samples

20% urethane was injected intraperitoneally at the dose of 8ml/kg body weight. After the rats were completely anesthetized, the eyeballs were removed, and the blood was taken. The blood was left at room temperature for 20 minutes, and centrifuged at 4°C for 3000 rpm for 25 minutes. The upper serum was sucked and stored at—20°C for later use. The concentration of serum $Mg^{2+}$ is detected by a biochemical Kit (Jiancheng, Nanjing, China). The levels of NO, ET-1 and ROS in serum are detected by NO assay kit (Nitrate reductase method), ET-1 Assay Kit and ROS Assay Kit (all Kits were purchased from Jiancheng, Nanjing, China), respectively. The levels of malondialdehyde (MDA), tumor necrosis factor-α (TNF-α) and Interleukin-1 (IL-1β) in serum were collectively tested using ELISA kits (SenBeiJia, Nanjing, China) according to the manufactures' protocol.

### Vascular function

SD rats were anesthetized by intraperitoneal injections of 20% urethane (8ml/kg) 24 h after the last exhaustive swimming session, and then sacrificed by cutting the femoral artery, resulting in exsanguination. The thoracic aorta was removed quickly and carefully, and then placed in the cold and oxygenated modified kreb's solution which contains 118 mmol/L NaCl, 4.7 mmol/L KCl, 1.18 mmol/L $KH_2PO_4$, 25 mmol/L $NaHCO_3$, 1.2 mmol/L $MgSO_4$, 10 mmol/L glucose and 2 mmol/L $CaCl_2$ (pH 7.4) [24]. Removed adipose tissue around blood vessels, and then cut it into rings, each 3 mm long. Thoracic aorta rings with endothelium were then mounted in a wire-myograph system (model 630 MA; Danish Myo Technology A/S, Aarhus, Denmark) using two stainless-steel wires. The chamber was filled with modified Kreb's solution which was gassed with 95% $O_2$ and 5% $CO_2$ continuously and maintained at 37°C and pH 7.4. All thoracic aorta rings initially were stretched to an optimal resting tension of 10 mN and allowed to equilibrate for 1hour. During this period, the modified Kreb's solution was changed every 15 min. Then, the rings were contracted with two 15min exposures to 60 mM KCl to obtain a reference contraction and to ensure smooth muscle viability. The rings then were washed with control solution for 3 times, and pre-contracted with $10^{-6}$ M phenylephrine (PE), and when the contraction curve tends to be stable, $10^{-5}$ M acetylcholine (ACh) was added to assess the integrity of the endothelium. The active tension caused by the agonist was normalized to the maximal contraction generated by 60 mM KCl.

### Experimental protocols for arteries

Thoracic aorta rings with endothelium were incubated for 1 h in Krebs solution that was contains normal (1.2 mM) or high (4.8 mM) magnesium concentrations, before and during the applied protocols. During this period, the modified Kreb's solution was changed every 15 min.

## Statistical analysis

Curve fitting was performed using SigmaPlot 11.0 software (Systat Software, Inc, Chicago, IL). All data are presented as means ± SE. Depending on the normality, the effects of exhaustive exercise on thoracic aortic thickness, Serum $Mg^{2+}$, Il-1β, TNF-α, ROS and MDA were analyzed by t-test or one-way ANOVA wherever applicable. Due to the factorial design of the study, the effects of exhaustive exercise and $Mg^{2+}$ on mice were analyzed by two-way ANOVA. With the aim of measuring the effect size of exhaustive exercise, Cohen's d were calculated. $P < 0.05$ means significant difference and $P < 0.01$ means extremely significant difference.

## Results

### Effect of exhaustive exercise on the morphology of thoracic aorta and serum $Mg^{2+}$ in SD rats

After 4-week exhaustive exercise intervention, we used HE staining to observe the potential effect of exhaustive exercise on the morphology of thoracic aorta in SD rats (Fig 1A). The morphological data showed that the intima of the thoracic aorta in the CON group was normal. The surface of endothelium was smooth and intact, and there was no protrusion or defect in vascular intima. The smooth muscle cells in the middle were arranged orderly (Fig 1A left). While EEE rats exhibited a significantly increase in aortic thickness and vascular smooth muscle cells are arranged disorderly, and the arterial intima structure is disordered, the continuity of the endothelium is interrupted (Fig 1A right). The ratio of thoracic aortic thickness was increased in EEE group (CON: 5.15 ± 0.11%, n = 18; EEE: 6.25 ± 0.20%, n = 12, $P < 0.01$, d = -1.88) (Fig 1B). Serum $Mg^{2+}$ concentration was significantly decreased in EEE groups (CON: 0.84 ± 0.03mM, n = 11; EEE: 0.56 ± 0.05, n = 10, $P < 0.01$, d = 2.21) (Fig 1C).

### Effects of exhaustive exercise on serum Il-1β and TNF-α in SD rats

After 4-week of exhaustive exercise intervention in SD rats, compared with CON rats, the levels of IL-1β (CON: 7.32 ± 0.60 ng/L, n = 6; EEE: 8.54 ± 0.97 ng/L, n = 8, $P < 0.05$, d = -1.52) and TNF- α (CON: 47.37 ± 4.89 ng/L, n = 8; EEE: 59.86 ± 6.00 ng/L, n = 10, $P < 0.01$, d = -2.29) in the serum of EEE rats were all significantly increased (Fig 2). The results showed that exhaustive exercise could promote the formation of chronic inflammation.

### Effects of exhaustive exercise on serum ROS and MDA in SD rats

Under normal physiological conditions, the production and elimination of ROS in the body are in dynamic balance. ROS at physiological level is an essential substance for producing muscle strength, maintenance of muscle content, gene expression, intracellular signal transduction and other related activities [25]. When the production rate of ROS in the body is much higher than the clearance rate during exercise, it will damage the tissues and lead to the decline of body function [26]. Compared with CON group, the serum ROS content of SD rats in EEE group was significantly increased (ROS fluorescence value: CON: 239.26 ± 31.07, n = 8; EEE: 336.54 ± 36.50, n = 10, $P < 0.01$, d = -2.87) (Fig 3A). Malondialdehyde (MDA) concentration is a biomarker of lipid peroxidation. Studies have shown that oxidative stress leads to lipid peroxidation, leading to the formation of harmful products of MDA, which can objectively reflect the level of free radicals in vivo [27]. Exhaustive exercise could significantly increase the content of MDA (CON: 6.34 ± 0.54 μmol/L, n = 8; EEE: 8.40±1.38 μmol/L, n = 10, $P < 0.01$, d = -1.96) (Fig 3B).

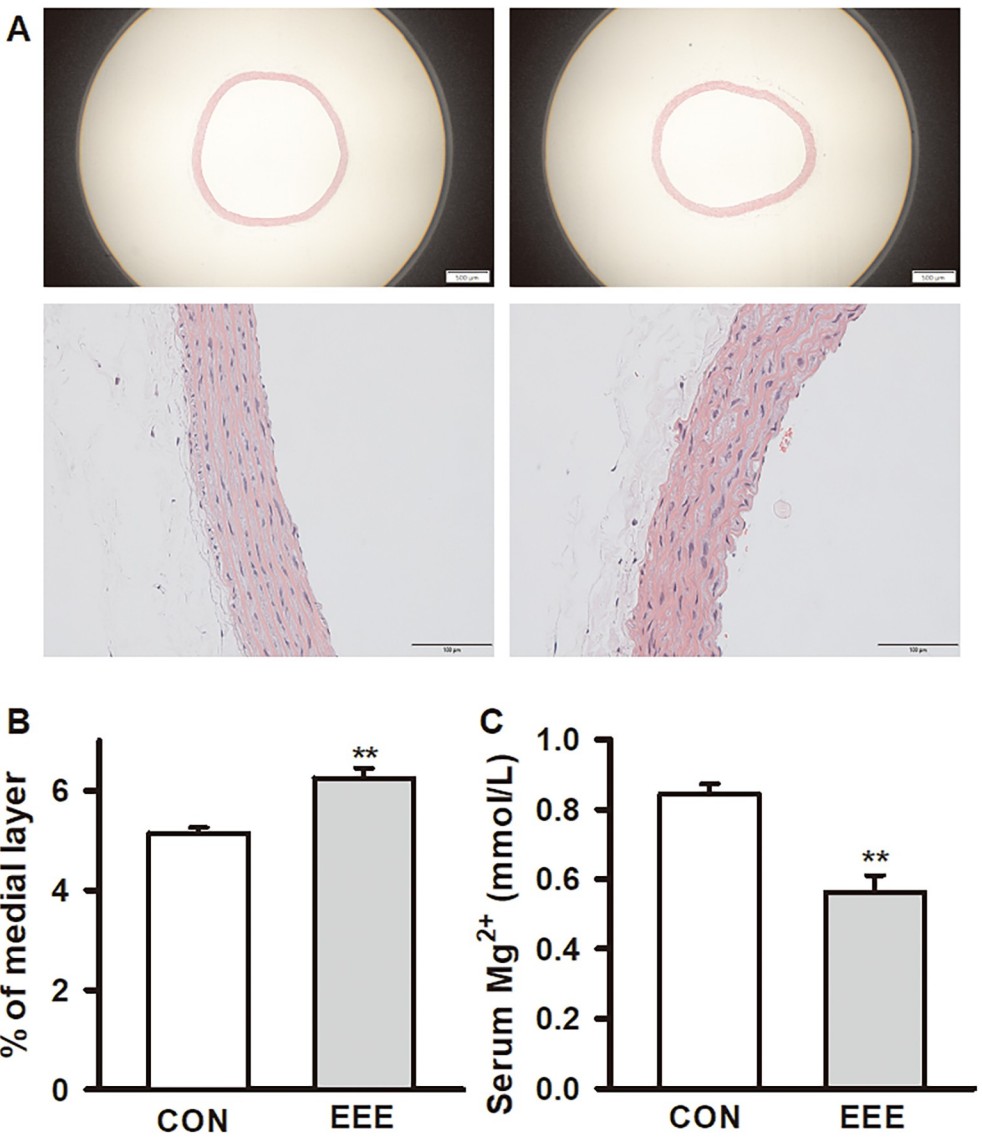

**Fig 1. Effects of exhaustive exercise on morphology of thoracic aorta and serum Mg$^{2+}$ in SD rats.** (A) left pictures are representative images of CON group taken at 40× magnification and 200× magnification, respectively and right pictures are representative images of EEE group taken at 40× magnification and 200× magnification, respectively. (B) The ratio of thoracic aortic thickness in CON and EEE groups. (C) The concentration of serum Mg$^{2+}$ in each group. Data presented as mean ± SE. $^{*}P < 0.05$ and $^{**}P < 0.01$ compared with the CON group.

## Effect of exhaustive exercise on serum NO and ET-1 in SD rats

NO is a crucial vasodilator synthesized and released by vascular endothelial cells. NO is a key molecule of vascular homeostasis, and its abnormal release is closely related to the occurrence and development of vascular disease [28]. NO is also an important signaling molecule involved in multiple physiological and pathophysiological cardiovascular [29]. Compared with CON group, the serum NO content in EEE group was significantly decreased (CON: 14.43 ± 2.16 μmol / L, n = 8; EEE: 9.02 ± 0.89 μmol / L, n = 5, $P < 0.01$, d = 3.28) (Fig 4A), and the ratio of NO/ET-1 was significantly decreased (CON: 0.48 ± 0.09, n = 8; EEE: 0.32 ± 0.05, n = 5, $P < 0.01$, d = 2.31) (Fig 4C).

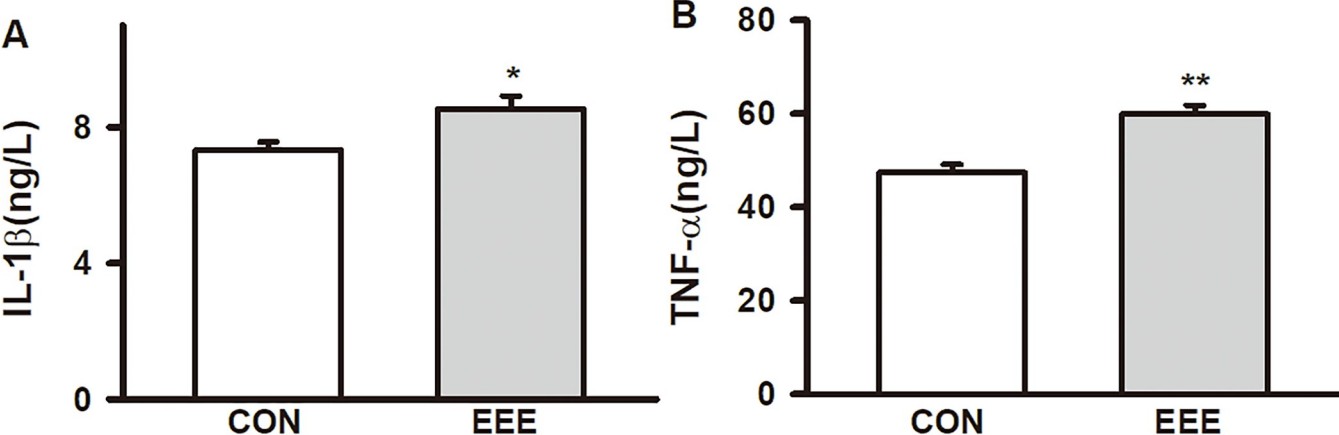

**Fig 2. Effects of exhaustive exercise on serum Il-1β and TNF-α in SD rats.** The concentration of IL-1β and TNF-α detected in serum from two groups. Data presented as mean ± SE. *$P < 0.05$ and **$P < 0.01$ compared with the CON group.

## Effect of magnesium ion on contractile function of thoracic aorta in EEE rats

KCl (60mM) was added to the aortic ring to make the thoracic aorta precontract. Compared with the CON group (normalized), the contractile response of isolated aorta in EEE group increased, and in high magnesium solution (4.8 mM), the maximum contraction of isolated aorta in EEE group induced by KCl was decreased, with exercise effect [F (1,114) = 45.883, P < .001] and Mg$^{2+}$ effect [F (1,114) = 93.2112, P < .001] (Fig 5E). These results showed that exhaustive exercise increased the contraction of thoracic aorta induced by KCl, and the increase of magnesium concentration improved the contraction function of thoracic aorta in exhausted rats.

## Effects of magnesium on ACh-induced concentration-dependent relaxation of thoracic aortic with intact endothelium from EEE rats

When acetylcholine (ACh: 10$^{-5}$M) was added to the aortic rings pre-contracted by phenylephrine (phen: 10$^{-6}$M), compared with the CON group, the diastolic response of isolated aorta in

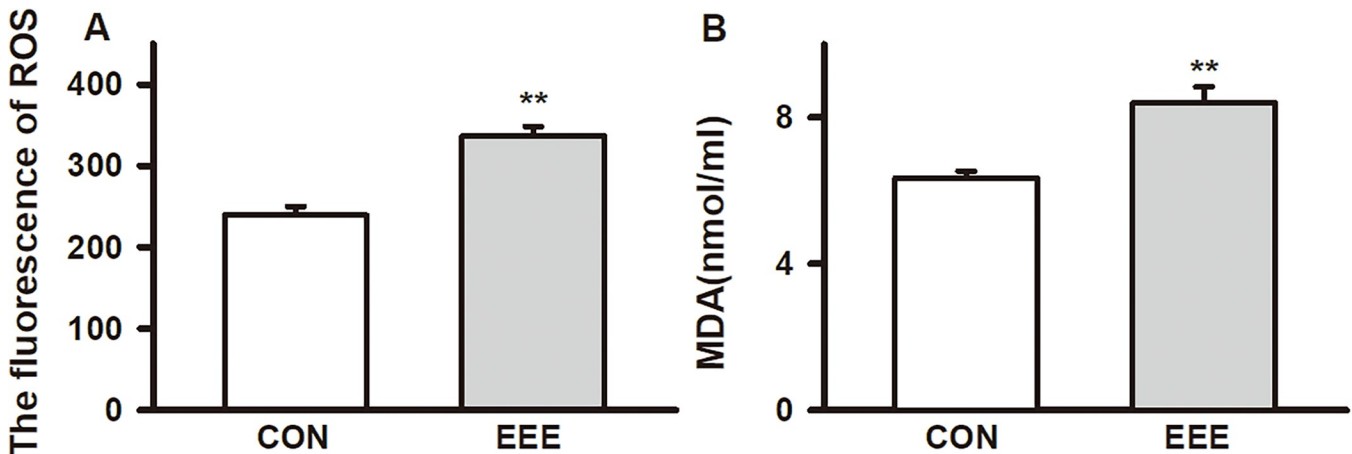

**Fig 3. Effects of exhaustive exercise on serum ROS and MDA in SD rats.** The concentration of ROS and MDA detected in serum from two groups. Data presented as mean ± SE. *$P < 0.05$ and **$P < 0.01$ compared with the CON group.

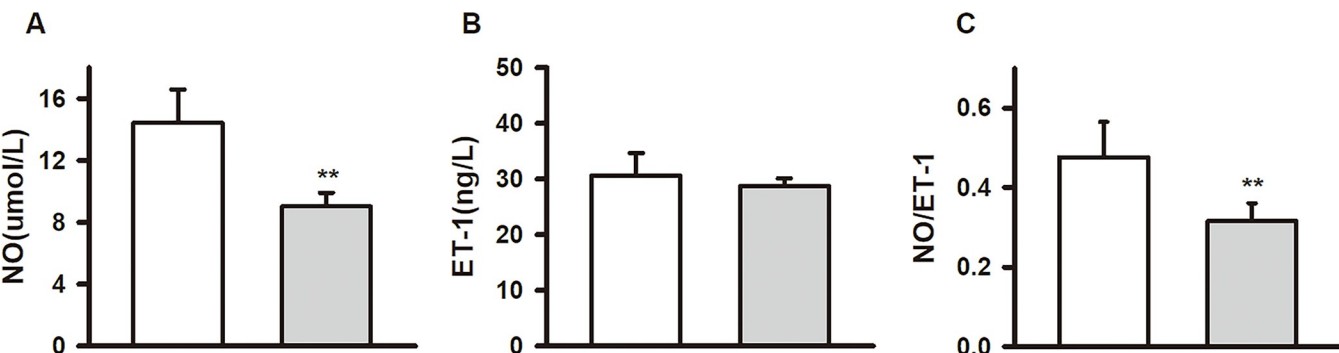

**Fig 4. Effects of exhaustive exercise on serum NO and ET-1 in SD rats.** (A) and (B)The concentration of NO and ET-1 detected in serum from two groups. (C) The results of the ratio of NO/ET-1 in the two groups. Data presented as mean ± SE. $^*P < 0.05$ and $^{**}P < 0.01$ compared with the CON group.

the EEE group was attenuated, and in high magnesium solution (4.8 mM), the maximum ACh induced relaxation of isolated aortas in EEE group was increased, with exercise effect [F (1,80) = 56.50, P < .001] and $Mg^{2+}$ effect [F (1,80) = 13.32, P < .001] (Fig 6E). These results showed that exhaustive exercise attenuated ACh induced endothelium-dependent relaxation of thoracic aorta, while the increase of magnesium concentration improved the endothelium-dependent relaxation of thoracic aorta in exhausted rats.

## Discussion

Exhaustive exercise can lead to impaired vascular function, while magnesium ion can improve vascular function. Therefore, we explored the protective effect of magnesium ion in exhausted rats in this study. Our major findings are the following: (1) 4-week exhaustive swimming exercise resulted in disordered structure and continuity of the vascular endothelial cell layer in rats, and the media was thickened, suggesting that 4-week exhaustive swimming exercise can lead to endothelial damage and smooth muscle proliferation; (2) 4-week exhaustive swimming exercise can significantly decrease the levels of serum $Mg^{2+}$; (3) 4-week exhaustive swimming exercise can significantly increase the levels of serum ROS, MDA, Il-1β and TNF-α in rats; (4) 4-week exhaustive swimming exercise can lead to a significant decrease in serum NO and a significant increase in the ratio of ET-1/NO in rats; (5) 4-week exhaustive swimming exercise can significantly increase the vasoconstriction induced by KCl and decrease the vasodilation induced by Ach, and high $Mg^{2+}$ can significantly improve the vasoconstriction KCl-induced and vasodilation Ach-induced. Hence our results, we emphasized the effects of $Mg^{2+}$ on vascular function in exhaustion exercise mice, and $Mg^{2+}$ can significantly improve vasomotor function in exhausted rats.

Blood vessels generally include endothelial cell layer, smooth muscle layer, and adipose outer membrane layer. Vascular endothelial cells are contact with blood directly and are the natural barrier between blood and vascular tissue [30]. Vascular smooth muscle cells (VSMCs) are the main components of the blood vessel wall, which alternate with collagen fibers and elastic fibers to form the intermediate membrane of blood vessels. After being regulated by nerves and humors, VSMCs generate tension through the interaction of myosin and actin in their cells, resulting in the contraction and relaxation of vascular smooth muscle cells [31]. Compared with the control group, the 4-week exhaustive swimming exercise resulted in disordered structure and continuity of the vascular endothelial cell layer in rats, and the media was thickened, so it has suggested that 4-week exhaustive swimming exercise could lead to vascular endothelial injury and smooth muscle proliferation. In addition, the results of this study showed that, 4-week exhaustive

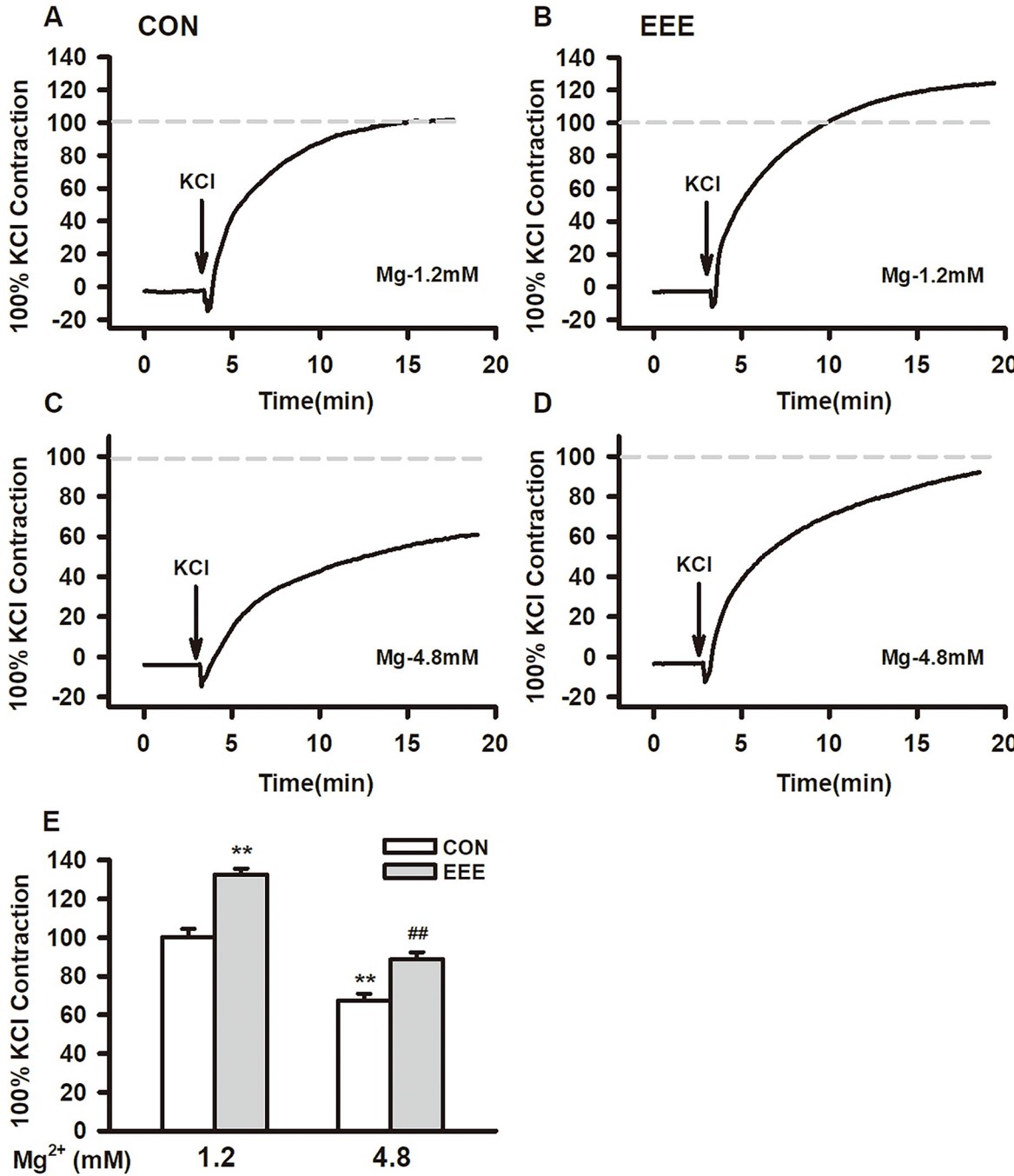

**Fig 5. The effect of magnesium on the KCl-induced contraction response in thoracic aortic from two group rats.** (A–D) Typical traces showing KCl -induced contraction responses in thoracic aortic at 1.2 and 4.8 mM magnesium, respectively. (E) Bar graphs showing the average values of the maximal contraction. Data are expressed as percentages of the 60 mM $K^+$-induced contractile response in con treated by 1.2mM magnesium. $**P < 0.01$ or $##P < 0.01$ compared with 1.2 mM magnesium in each group. Data are presented as means ± SE.

swimming exercise resulted in enhanced vasoconstriction induced by KCl and decrease the vaso-dilation induced by Ach, further suggesting that 4-week exhaustive swimming exercise induced vascular impaired smooth muscle cell and vascular endothelial cell function.

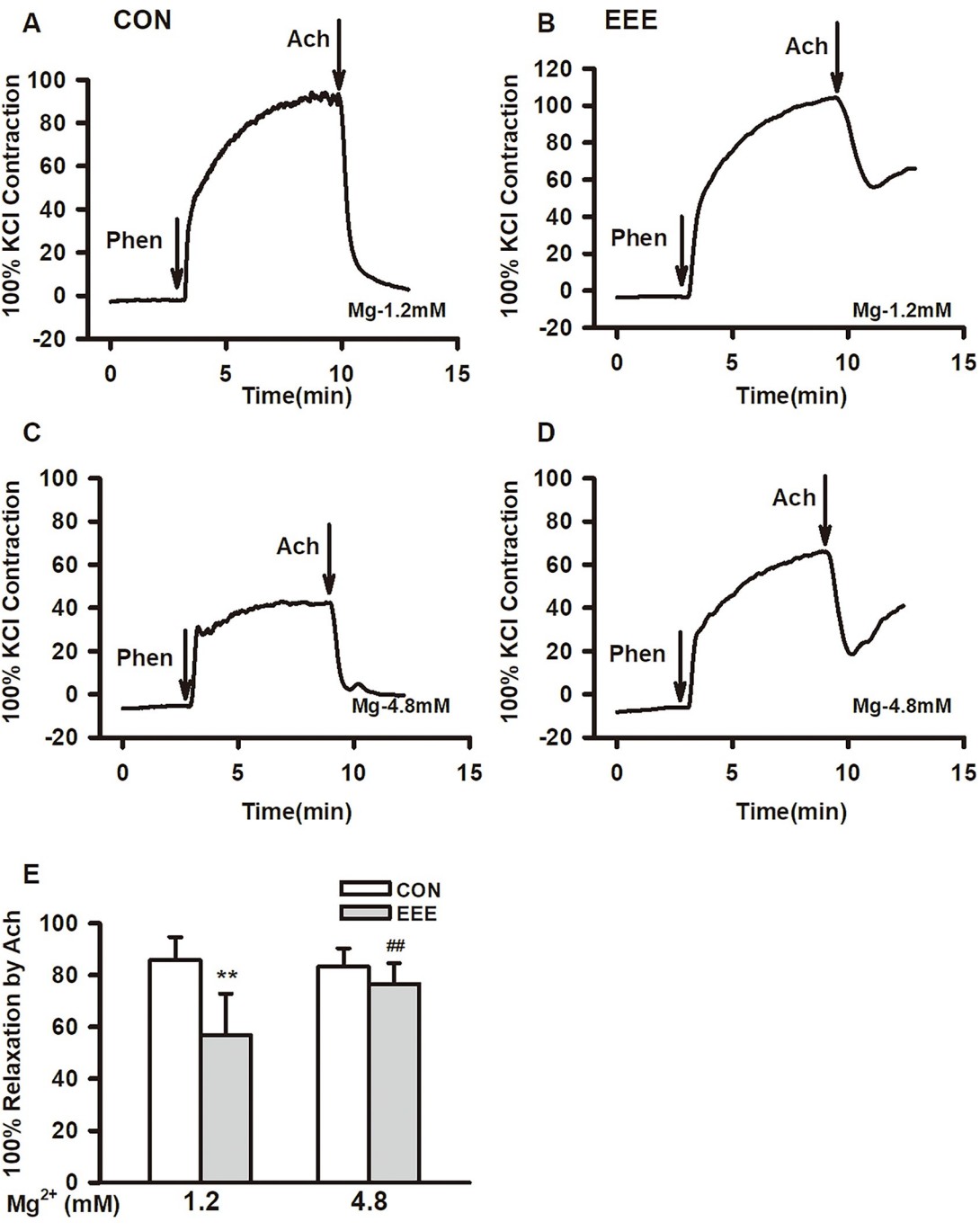

**Fig 6. The effect of magnesium on the acetylcholine (ACh)-induced relaxant response in thoracic aortic from two group rats.** (A–D) Typical traces showing ACh-induced relaxant responses in thoracic aortic at 1.2 and 4.8 mM magnesium, respectively. (E) Bar graphs showing the average values of the maximal Relaxation. Abbreviation: Phen, phenylephrine. $^{**}P < 0.01$ or $^{\#\#}P < 0.01$ compared with 1.2 mM magnesium in each group. Data are presented as means ± SE.

A large number of studies have shown that regular or appropriate exercise will produce low levels of ROS, and excessive endogenous free radicals produced during exercise can damage the physiological functions of the whole tissue [25, 32, 33]. It's demonstrated that a moderate-intensity exercise program could be beneficial to attenuating the susceptibility of oxidative

damage and to increasing endothelial function [34]. Our previous research also proved that aerobic exercise decreased the elevated MDA while increased the reduced SOD in IGT mice [35]. Long term strenuous exercise can cause a significant increase in oxygen uptake which may lead to an increase in the body's metabolic activity and oxygen utilization, leading an increase of ROS production. And at the same time, with the increase of superoxide anion in the cytoplasm, the superoxide anion can react with NO to produce peroxynitrite (ONOO -), which will lead to the decoupling of endothelial NO synthase (eNOS) and further produce superoxide anion, aggravating the oxidative stress reaction, leading to inflammatory reaction [36]. An imbalance between prooxidants and antioxidants has been implicated in cardiovascular and metabolic diseases [37]. Oxidative stress may lead to subsequent oxidative modification or damage to lipids, proteins, and DNA, with deleterious consequences for metabolic and vascular disease [25, 38, 39]. Research have demonstrated that aerobic exercise could reduce the level of serum TNF-$\alpha$ and IL-1$\beta$, and increase the concentration of anti-inflammatory factor IL-10 in high-fat diet rats [40]. Long term high-intensity exercise can cause systemic low-grade inflammation. The reason may be that long-term strenuous exercise increases the blood flow supply of muscles and causes insufficient perfusion of internal organs. In addition, sports physical trauma may lead to ischemia and physical injury in sports organ, thus causing systemic inflammatory response [41]. The concentration of inflammatory factors in blood can reflect the inflammatory state of blood vessels. Research have shown that chronic exhaustive exercise significantly increased the serum levels of pro-inflammatory cytokines (IL-1$\beta$, TNF-$\alpha$, IFN-$\gamma$) and the IFN-$\gamma$/IL-4 ratio, and decreased the anti-inflammatory cytokines (IL-4, IL-10) [42]. Consistent with this, our study found that 4-week exhaustive swimming exercise can significantly increase the levels of serum Il-1$\beta$ and TNF-$\alpha$ in rats, and the body was in an inflammatory state.

Vascular endothelium is the primary site of dysfunction in metabolic and cardiovascular diseases. Endothelial cells are endocrine organs, which can actively regulate vasocontraction and relaxation by releasing some chemical substances. In addition, endothelial cells are also important in controlling leukocyte activation, platelet adhesion, aggregation and migration, and play major roles in the regulation of inflammatory response and angiogenesis [30]. Vascular endothelial dysfunction caused by the loss of NO bioavailability is one of the most important contributors to the pathogenesis of vascular diseases [43]. Generating sufficient NO levels to regulate the resistance of the blood vessels and thus maintain adequate blood flow is essential to the healthy performance of the vasculature. It has been reported that exercise training can enhance endothelium-dependent vasodilation in animal and human studies [44–47]. In general, regular exercise not only increases the production and release of the NO induced by shear stress, but also reduces NO inactivation by upregulating the antioxidant defense mechanism, ultimately leading to an increase in NO bioavailability [48]. Habitual aerobic exercise reduces arterial stiffness, and its underlying mechanism is associated with vasodilation by NO production via activation of eNOS phosphorylation in the arteries [49, 50]. Exhaustive exercise causes high-frequency and repeated blood flow shear stress, which leads to produce numerous free radicals, then the redox balance is destroyed [51, 52]. The key rate-limiting enzyme eNOS for NO synthesis is reduced, and the expression of eNOS-mRNA is down-regulated, finally the amount of NO production is reduced [53]. NO inhibits monocyte adhesion to endothelial cells, smooth muscle cell proliferation, and platelet aggregation, so depletion of NO causes endothelial dysfunction and abnormal vascular remodeling [54]. A study reported that the maximal relaxation of the aorta in response to Ach was decreased immediately after high-intensity swimming [55]. The results of our study have shown that 4-week exhaustive swimming exercise can cause a significant decrease in serum NO and a meaningful increase in the ratio of ET-1/NO in rats, suggesting that 4-week exhaustive swimming exercise leads to

endocrine dysfunction of vascular endothelial cells and smooth muscle cell proliferation, resulting in vascular dysfunction.

$Mg^{2+}$ can regulate cardiovascular function, such as regulating endothelial cell function, vascular smooth muscle tone and myocardial excitability, thus playing an essential role in some pathological changes in the cardiovascular system [56]. In the present study, magnesium exhibited dual effects on the KCl-induced thoracic aorta contraction, but a high concentration of magnesium attenuated the KCl -induced response. High magnesium has been reported improve endothelial function, whereas low magnesium can lead to endothelial dysfunction [57, 58]. In this study, we found the serum $Mg^{2+}$ was sharply decreased in EEE group, and this can damage the endothelium. Chronic inflammation and redox homeostasis occur in the body after long-term exhaustion, and lead to increase body calcium signal, which further impairs the functions of vascular endothelial cells and vascular smooth cells [59, 60]. As a $Ca^{2+}$ antagonist, a high concentration of extracellular magnesium can inhibit $Ca^{2+}$ entry. Extracellular $Mg^{2+}$ can bind to the fixed negative charges on the outer surface of the cell membrane, resulting in electrostatic shielding, causing an increase in the activation threshold of voltage gated $Ca^{2+}$ channels [61]. High extracellular $Mg^{2+}$ can also reduce $Ca^{2+}$ influx by competing with $Ca^{2+}$ for the pores of $Ca^{2+}$ channels, resulting in physical blocking or allosteric modulation of the channels [62]. Increased $[Mg^{2+}]_i$ can inhibit $Ca^{2+}$ release from sarcoplasmic reticulum inositol triphosphate receptors and activate $Ca^{2+}$-ATPase located in sarcoplasmic reticulum to enhance $Ca^{2+}$ chelation [63]. In addition to its effects on vasoconstriction, magnesium can also improve vasodilatation by modulating endothelial function. The vascular endothelium plays an important role in regulating vasomotor tone by synthesizing and releasing prostacyclin and NO [64]. Previous studies have shown that the relaxant action of magnesium was mediated by the endothelial release of NO [65, 66]. Moreover, $Mg^{2+}$ can inhibit the depolarization effect of $Ca^{2+}$ and the excitation-contraction coupling related to $Ca^{2+}$, so $Mg^{2+}$ can promote vascular relaxation of blood vessels by inhibiting $Ca^{2+}$ channels and their functions [67, 68]. All these could result in the attenuation thoracic aorta contraction and increasing vasodilation at a high magnesium concentration.

To sum up, the results of the present study showing 4-week exhaustive swimming exercise increases the oxidative stress of the body and lead to chronic inflammation, meanwhile the serum $Mg^{2+}$ decreased in EEE group, which damages the vascular intima, thickens the vascular smooth muscle layer, and result in vascular dysfunction, while high extracellular $Mg^{2+}$ can significantly improve the vasomotor function of exhaustive exercise rats, so $Mg^{2+}$ can significantly improve vasomotor function in exhausted rats. This will provide a new preventive measure for vascular injury caused by exhaustive exercise, and magnesium can be appropriately supplemented during long-term high-intensity exercise.

## Limitations and perspective

The high magnesium experiment has not been studied in vivo, which is the limitation of the study. The current research shows that magnesium ion has a protective effect on vascular. Our research shows that high magnesium can improve the vasomotor function in exhausted rats, but no possible mechanism has been explored. The specific mechanism can be further explored by high magnesium experiment in vivo in the future.

## Supporting information

**S1 Dataset.**
(DOCX)

## Acknowledgments

The authors thank the laboratory team for their technical assistance.

## Author Contributions

**Data curation:** Zong-Xiang Li.

**Formal analysis:** Zong-Xiang Li, Yan-Zhong Liu.

**Funding acquisition:** Dan Wang, Yi-Ping Liu.

**Methodology:** Dong-Mou Jiang.

**Software:** Xin Wang.

**Supervision:** Yi-Ping Liu.

**Writing – original draft:** Dan Wang.

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
