## [Decision Letter · Decision Letter 0]

11 Oct 2022

PONE-D-22-24419The Role of Magnesium Ions in the Improvement of Vascular Function in Exhausted RatsPLOS ONE

Dear Dr. Wang,

Thank you for submitting your manuscript to PLOS ONE. After careful consideration, we feel that it has merit but does not fully meet PLOS ONE’s publication criteria as it currently stands. Therefore, we invite you to submit a revised version of the manuscript that addresses the points raised during the review process. It is advised that results on the effect of moderate exercise on the mediators analyzed with that of the observations made in this study are to be included. Bibliography included should be as latest as possible. Statistical analyses are to be redone for better clarity. Provide strong rationale for the study in the introduction. The limitations of the study should be mentioned in the discussion.

We look forward to receiving your revised manuscript.

Kind regards,

Suresh Yenugu

Academic Editor

PLOS ONE

Journal Requirements:

Reviewers' comments:

Reviewer's Responses to Questions

**Comments to the Author**

1. Is the manuscript technically sound, and do the data support the conclusions?

Reviewer #1: Yes

Reviewer #2: Yes

2. Has the statistical analysis been performed appropriately and rigorously? 

Reviewer #1: Yes

Reviewer #2: No

3. Have the authors made all data underlying the findings in their manuscript fully available?

Reviewer #1: Yes

Reviewer #2: Yes

4. Is the manuscript presented in an intelligible fashion and written in standard English?

Reviewer #1: Yes

Reviewer #2: Yes

5. Review Comments to the Author

Reviewer #1: The manuscript is technically sound and presents scientifically relevant data. However, some points need to be addressed. The authors compare control rats to EEE rats and show an increase in interleukin-1 β (IL-1β), tumor necrosis factor –α (TNF-α), malondialdehyde (MDA), reactive oxygen species (ROS) in serum. A decrease in reported in the nitric oxide (NO) and endothelin 1 (ET-1) ratio. However, there is no comparison to moderate exercise, which the authors mention has been inversely linked to mortality. Therefore, the concentrations of these mediators in moderate exercise conditions need to mentioned at least in discussion for a clear understanding of this study.

Minor changes:

1. Page 7: change 'scientifical regular physical activity'.

2. Page 7: 'exercise time exceeding' to 'exercise time exceed'.

3. Page 7: 'making the body in a state of' to 'triggering'

and similar language changes in rest of the text are required.

Reviewer #2: This study aimed to evaluate the effects of Mg2+ on vascular function damage caused by a long-term exhaustive exercise.

According to the authors, the long-term exhaustive exercise induced an increase on oxidative stress, leading to inflammation, while Mg2+ can significantly improve the vasomotor function of exhaustive exercised rats.

The study is well organized and written. The methods used are all well-accepted and described. Results are presented in a logical manner, which facilitates the comprehension of the paper. The finds will certainly contribute to the academic area.

Please find below some suggestions and comments in the attempt to contribute to the quality of the manuscript:

1- In my opinion the title should be more specific, addressing directly the main find of the manuscript;

2- The references used in the manuscript are appropriate. However, only 21 from all 61 references used (~34%) are from the last 5 years;

3- A clear stated hypothesis would benefit the manuscript. I would suggest authors include a hypothesis at the end of the Introduction, and discuss if it was confirmed or not at the beginning of the Conclusion section;

4- The discussion section should start by presenting the main find of the study;

5- Regarding statistical analysis, I understand that the use of independent t-test is appropriate, however, the one-way ANOVA (described in methods) is not. Considering the results presented on figures 5E and 6E, we can observe the presence of 2 factors (groups and Mg2+ concentration), what makes necessary the use of a two-way ANOVA;

6- Also, report magnitude regarding statistical significance is not recommended (“P < 0.05 means significant difference and P < 0.01 means extremely significant difference”; page 12). I would recommend authors to calculate and provide the effect sizes (Cohen’s d or Hedges’ g) of each comparison, which can provide a greater value for understanding the magnitude of the observed changes;

7- Authors have shown an increase of IL-1beta and TNF-alfa serum concentration. A balance between pro- and anti-inflammatory cytokines is essential to understand the real inflammatory state induced by a specific treatment. Is it possible to evaluate some anti-inflammatory cytokines? If not, please consider use the literature to discuss this topic more detailed in the discussion;

8- Another topic that could be considered in the discussion is the known reaction that occurs between anion superoxide and nitric oxide (forming peroxynitrite), which is much faster than the dismutation of superoxide by the antioxidant enzyme SOD. Considering the participation of peroxynitrite in oxidative stress and inflammation, combined to the results obtained in this study, I would recommend authors to consider add some discussion regarding this process;

9- Does this study have any limitations? It would be good to have it described in the discussion. Some “directions” for further studies should be also added.

6. PLOS authors have the option to publish the peer review history of their article (what does this mean?). If published, this will include your full peer review and any attached files.

Reviewer #1: No

Reviewer #2: No

---

## [Author Response · Author response to Decision Letter 0]

14 Nov 2022

We would like to thank the reviewer for the positive comments, as well as for the constructive suggestions. We have revised the manuscript as required, and believe these changes have significantly enhanced the quality of our paper. The followings are our responses to the reviewer’s comments.

Response to Reviewer #1:

 The manuscript is technically sound and presents scientifically relevant data. However, some points need to be addressed. The authors compare control rats to EEE rats and show an increase in interleukin-1 β (IL-1β), tumor necrosis factor –α (TNF-α), malondialdehyde (MDA), reactive oxygen species (ROS) in serum. A decrease in reported in the nitric oxide (NO) and endothelin 1 (ET-1) ratio. However, there is no comparison to moderate exercise, which the authors mention has been inversely linked to mortality. Therefore, the concentrations of these mediators in moderate exercise conditions need to mention at least in discussion for a clear understanding of this study.

We would like to thank the reviewer for the positive comments, as well as for the constructive suggestions. We have revised the manuscript according to your comments, and the effects of moderate intensity exercise on the changes of serum interleukin-1 β (IL-1β), tumor necrosis factor –α (TNF-α), malondialdehyde (MDA), reactive oxygen species (ROS) were described separately in the discussion. We believe these changes have significantly enhanced the quality of our paper. The followings are the point-to-point reply to your comments.

 [1. Page 7: change 'scientifical regular physical activity'.]

Response：We sincerely thank you for the comment. We have changed 'scientifical regular physical activity' to ' physical activity' in the revision.

[2. Page 7: 'exercise time exceeding' to 'exercise time exceed'.]

Response：Thank you for your suggestion. We have changed 'exercise time exceeding' to 'exercise time exceed' in the revised manuscript.

[3. Page 7: 'making the body in a state of' to 'triggering' and similar language changes in rest of the text are required.]

Response：Thank you very much for your friendly suggestion. We have revised accordingly. And we revised the whole manuscript carefully to avoid language errors. In addition, we asked several colleagues who are native English speakers to check the English. We believe that the language is now acceptable for the review process.

Response to Reviewer #2:

This study aimed to evaluate the effects of Mg2+ on vascular function damage caused by a long-term exhaustive exercise. According to the authors, the long-term exhaustive exercise induced an increase on oxidative stress, leading to inflammation, while Mg2+ can significantly improve the vasomotor function of exhaustive exercised rats. The study is well organized and written. The methods used are all well-accepted and described. Results are presented in a logical manner, which facilitates the comprehension of the paper. The finds will certainly contribute to the academic area. Please find below some suggestions and comments in the attempt to contribute to the quality of the manuscript.

We would like to thank the reviewer for the positive comments, as well as for the constructive suggestions. We have revised the manuscript according to your comments. We believe these changes have significantly enhanced the quality of our paper. The followings are the point-to-point reply to your comments.

[1- In my opinion the title should be more specific, addressing directly the main find of the manuscript.]

Response：Thank you for your suggestion. We have changed the title to “Magnesium Ions Improve Vasomotor Function in Exhausted Rats” as suggested.

[2- The references used in the manuscript are appropriate. However, only 21 from all 61 references used (~34%) are from the last 5 years.]

Response：Thank you very much for the suggestion. We have updated the references and retained several high-level and necessary references, and now there are 44 of all 68 references used (~65%) from the last 5 years.

[3- A clear stated hypothesis would benefit the manuscript. I would suggest authors include a hypothesis at the end of the Introduction, and discuss if it was confirmed or not at the beginning of the Conclusion section.]

Response：We sincerely thank the reviewer for the comment. We have added a hypothesis at the end of the Introduction and confirmed it at the beginning of the Conclusion section.

[4- The discussion section should start by presenting the main find of the study.]

Response：Thank you very much for the suggestion. At the beginning of the discussion, we highlighted the role of magnesium ions and summarized our main findings by presenting the results, and discussed step by step.

[5- Regarding statistical analysis, I understand that the use of independent t-test is appropriate, however, the one-way ANOVA (described in methods) is not. Considering the results presented on figures 5E and 6E, we can observe the presence of 2 factors (groups and Mg2+ concentration), what makes necessary the use of a two-way ANOVA.]

Response：Thank you for pointing out our error. The statistical method has been corrected in methodology, and Figures 5E and 6E are recounted using a two-way ANOVA.

[6- Also, report magnitude regarding statistical significance is not recommended (“P < 0.05 means significant difference and P < 0.01 means extremely significant difference”; page 12). I would recommend authors to calculate and provide the effect sizes (Cohen’s d or Hedges’ g) of each comparison, which can provide a greater value for understanding the magnitude of the observed changes.]

Response：Thank you very much for the suggestion. We have provided the effect sizes (Cohen’s d) of each comparison in the revised manuscript.

[7- Authors have shown an increase of IL-1beta and TNF-alfa serum concentration. A balance between pro- and anti-inflammatory cytokines is essential to understand the real inflammatory state induced by a specific treatment. Is it possible to evaluate some anti-inflammatory cytokines? If not, please consider use the literature to discuss this topic more detailed in the discussion.]

Response：We sincerely thank you for the comment. We have added the literature to understand the changes of anti-inflammatory factors during exhaustive exercise. The details are as follows: Research have shown that chronic exhaustive exercise significantly increased the serum levels of pro-inflammatory cytokines (IL-1β, TNF-α, IFN-γ) and the IFN-γ/IL-4 ratio, and decreased the anti-inflammatory cytokines (IL-4, IL-10) (Lu J et al. 2012).

[8- Another topic that could be considered in the discussion is the known reaction that occurs between anion superoxide and nitric oxide (forming peroxynitrite), which is much faster than the dismutation of superoxide by the antioxidant enzyme SOD. Considering the participation of peroxynitrite in oxidative stress and inflammation, combined to the results obtained in this study, I would recommend authors to consider add some discussion regarding this process.]

Response：Thank you for the insightful comments. We have added the discussion that peroxynitrite participates in oxidative stress and inflammation. The details are as follows: At the same time, with the increase of superoxide anion in the cytoplasm, the superoxide anion can react with NO to produce peroxynitrite (ONOO -), which will lead to the decoupling of endothelial NO synthase (eNOS) and further produce superoxide anion, aggravating the oxidative stress reaction, leading to inflammatory reaction (Xu C et al. 2016).

[9- Does this study have any limitations? It would be good to have it described in the discussion. Some “directions” for further studies should be also added.]

Response：Thank you for your insightful suggestion. We added limitations and prospective after the discussion. The details are as follows: The high magnesium experiment has not been studied in vivo, which is the limitation of the study. The current research shows that magnesium ion has a protective effect on vascular. Our research shows that high magnesium can improve the vasomotor function in exhausted rats, but no possible mechanism has been explored. The specific mechanism can be further explored by high magnesium experiment in vivo in the future.

---

## [Decision Letter · Decision Letter 1]

5 Dec 2022

Magnesium Ions Improve Vasomotor Function in Exhausted Rats

PONE-D-22-24419R1

Dear Dr. Wang,

We’re pleased to inform you that your manuscript has been judged scientifically suitable for publication and will be formally accepted for publication once it meets all outstanding technical requirements.

Kind regards,

Suresh Yenugu

Academic Editor

PLOS ONE

Additional Editor Comments (optional):

Reviewers' comments:

Reviewer's Responses to Questions

**Comments to the Author**

1. If the authors have adequately addressed your comments raised in a previous round of review and you feel that this manuscript is now acceptable for publication, you may indicate that here to bypass the “Comments to the Author” section, enter your conflict of interest statement in the “Confidential to Editor” section, and submit your "Accept" recommendation.

Reviewer #1: All comments have been addressed

Reviewer #2: All comments have been addressed

2. Is the manuscript technically sound, and do the data support the conclusions?

Reviewer #1: Yes

Reviewer #2: Yes

3. Has the statistical analysis been performed appropriately and rigorously? 

Reviewer #1: Yes

Reviewer #2: Yes

4. Have the authors made all data underlying the findings in their manuscript fully available?

Reviewer #1: Yes

Reviewer #2: Yes

5. Is the manuscript presented in an intelligible fashion and written in standard English?

Reviewer #1: Yes

Reviewer #2: Yes

6. Review Comments to the Author

Reviewer #1: My comments and criticisms have all been well addressed. I recommend the acceptance of this manuscript.

Reviewer #2: I congratulate the authors for this revised version of the manuscript. All my suggestions and comments have been adequately addressed.

7. PLOS authors have the option to publish the peer review history of their article (what does this mean?). If published, this will include your full peer review and any attached files.

Reviewer #1: No

Reviewer #2: **Yes: **Rafael Herling Lambertucci

---

## [Editor Report · Acceptance letter]

1 Feb 2023

PONE-D-22-24419R1 

Magnesium Ions Improve Vasomotor Function in Exhausted Rats 

Dear Dr. Wang:

I'm pleased to inform you that your manuscript has been deemed suitable for publication in PLOS ONE. Congratulations! Your manuscript is now with our production department. 

Kind regards, 

on behalf of

Dr. Suresh Yenugu 

Academic Editor

PLOS ONE